# Health Status and Activity Discomfort among Elderly Drivers: Reality of Health Awareness

**DOI:** 10.3390/healthcare11040563

**Published:** 2023-02-14

**Authors:** Sang-Hoon Jeong, Eun-Yeob Kim, Seung-Jin Lee, Woo-June Choi, Chilhwan Oh, Hwa-Jung Sung, Jaeyoung Kim

**Affiliations:** 1Research Institute for Skin Image, College of Medicine, Korea University, Seoul 08308, Republic of Korea; 2Core Research & Development Center, Korea University Ansan Hospital, Gyeonggi-do 15355, Republic of Korea; 3School of Electrical and Electronics Engineering, Chung-Ang University, Seoul 06974, Republic of Korea; 4Department of Dermatology, School of Medicine, Wonkwang University, Jeolabuk-do 54538, Republic of Korea; 5Department of Hematooncology, College of Medicine, Korea University, Seoul 02841, Republic of Korea

**Keywords:** elderly, driving risk, medical conditions, recognition, discomfort, health

## Abstract

As the number of elderly drivers rapidly increases worldwide, interest in the dangers of driving is growing as accidents rise. The purpose of this study was to conduct a statistical analysis of the driving risk factors of elderly drivers. In this analysis, data from the government organization’s open data were used for the secondary processing of 10,097 people. Of the 9990 respondents, 2168 were current drivers, 1552 were past drivers but were not driving presently, and 6270 did not have a driver’s license; the participants were divided into groups accordingly. The elderly drivers who were current drivers had a better subjective health status than those who were not. Visual and hearing aids were used in the current driving group, and their depression symptoms reduced as they drove. The elderly who were current drivers experienced difficulties while driving in terms of decreased vision, hearing loss, reduced arm/leg reaction speed, decreased judgment of the road conditions such as signals and intersections, and a decreased sense of speed. The results suggest that elderly drivers are unaware of the medical conditions that can negatively affect their driving. This study contributes to the safety management of elderly drivers by understanding their mental and physical status.

## 1. Introduction

Globally, the number of elderly drivers aged 65 and over was 7685 million in 2019, accounting for 14.9% of the total population [1]. This number is rapidly and continuously increasing. Therefore, as many accidents occur among elderly drivers, interest in the dangers of driving is increasing [2]. One study reported that elderly people with a driver’s license can improve their independence by self-driving: thus, increasing their autonomy in participating in old-age activities [3]. Among the elderly, driving is considered an essential action that expands the scope of activities, such as leisure activities, visits to hospitals, and shopping, and provides opportunities for independence in their daily lives [4]. In this way, elderly drivers have positive emotional and social functions, given the increasing opportunities for social activities [5]. Thus, elderly people who drive themselves are considered to have a relatively high level of life satisfaction [6]. Approximately 30,000 cases were reported in 2018 in the Republic of Korea, and this number is continuously increasing [7,8]. When accidents occur, the elderly suffer serious injuries and have a slow recovery rate compared to young people [2,4]. As such, elderly drivers have a high risk of traffic accidents, and their anxiety about accidents is severe compared to the other age groups [5,9].

The ability of elderly drivers to self-regulate changes in their driving ability by becoming more aware of and managing their health status is naturally strengthened with increasing age [10]. Nevertheless, the reliability of elderly drivers’ awareness of their health status and driving ability is controversial [11]. In countries such as the United States, the United Kingdom, Canada, and Australia, a self-reporting evaluation method was used to investigate the characteristics of elderly drivers [12,13]. Although they tend to avoid certain driving situations, such as night driving, long-distance driving, and driving when the roads are congested [8,14,15], they are affected by society and the culture to which they belong [16]. Some studies analyzed changes in behavior, cognition, perception, and physical function of elderly drivers while driving using the Self-report Assessment Forecasting Elderly Driving Risk (SAFE-DR), which was developed to assess the situation in the Republic of Korea [15,17,18].

Owing to medical advances and changes in the social environment, the proportion of elderly drivers is rapidly increasing and will continue to increase [1]. If elderly drivers are not aware of their physical changes and do not avail themselves of treatment in a timely manner, it interferes with their driving ability [5] and, consequently, increases the risk of accidents. This study aimed to analyze the physical characteristics, underlying diseases, and health consciousness of elderly drivers to identify their mental and physical conditions and help prevent traffic accidents. In addition, the researchers provide basic data for related research.

## 2. Materials and Methods

### 2.1. Study Design and Sampling

The data for this study were obtained from the Health and Welfare Data Portal of the Korea Institute of Health and Social Affairs and included the data of 10,097 elderly people in the Republic of Korea aged 65 years and over (National Statistics approval no. 117071). A total of 10,097 people were surveyed; 107 people who did not drive were excluded from the total, and the remaining 9990 people were divided into three groups: 2168 people who were currently driving, 1552 people who were past drivers but were not currently driving at the time of the survey, and 6270 people who had no driver’s license. Those with the highest age of elderly drivers at the time of the survey were selected and further classified as those without a driver’s license, past drivers, or not current drivers, who were at the time of the survey. The participants’ ages ranged from 65–90 years (Figure 1).

### 2.2. Data Variables

The data description of the variables used in this study is as follows: (1)Driving status, which was divided into two groups: past drivers (not currently driving) and not having a driver’s license.(2)Health status and health behavior, which included thoughts on health in general; presence of chronic diseases (diseases lasting for more than 3 months as diagnosed by a doctor, namely circulatory diseases: high blood pressure, stroke (stroke, cerebral infarction), hyperlipidemia (dyslipidemia), angina pectoris, and myocardial infarction (heart failure and arrhythmia); endocrinal disease: diabetes and thyroid disease; musculoskeletal diseases: osteoarthritis (degenerative arthritis), rheumatoid arthritis, osteoporosis, low back pain, sciatica, fracture, dislocation, and after effects of accidents; respiratory diseases: chronic bronchitis, emphysema, asthma, pulmonary tuberculosis, and tuberculosis, neuropsychiatric diseases: depression, dementia, Parkinson’s disease, and insomnia; sensory diseases: cataract, glaucoma, chronic otitis media, senile deafness, skin disease, and cancer (malignant neoplasm); digestive diseases: gastroduodenal ulcer, hepatitis, and liver cirrhosis; genitourinary diseases: chronic kidney disease, prostatic hyperplasia, urinary incontinence, and anemia, etc.(3)State of physical function, including eyesight (watching TV, reading newspapers), hearing (talking on the phone, talking to the person next to you), chewing (chewing meat or hard things), and determining muscle strength (active movement (running about one lap (400 m) on the playground), walking around the playground (400 m), climbing 10 steps without a break, bending over, squatting, or kneeling, and reaching out for something higher than one’s head). Physical functioning was divided into lifting, moving, and disability determination.(4)Depressive symptoms were measured using the shortened geriatric depression scale (SGDS)-K15, which is a Korean translation of the SGDS developed by [19] to evaluate depressive symptoms in the elderly population (out of a total score of 15, individuals with a score of 8 or higher were classified as having depressive symptoms).(5)Social activities and discomfort in social activities were classified into two categories, namely, difficulty in using the information necessary for life and the inconvenience caused by using information technology in everyday life.(6)Economic activity was classified into current income, work, and desired work.(7)Precognitive function: cognitive function was confirmed and measured using the Mini-Mental State Examination for Dementia Screening (MMSE-DS) test tool. A representative screening test developed by [20] is widely used for simple and rapid measurement as well as screening for any cognitive impairment; the standardized Korean version of the mini-mental state examination (MMSE-K) [21], the Korean mini-mental state examination (K-MMSE) [22], and the mini-mental state examination-Korean children (MMSE-KC) [23] have been used in the Republic of Korea. A total mini-mental state examination (MMSE) score of 30 points is considered the cut-off point for cognitive impairment; a score of 0–10 indicates severe cognitive impairment, 10–20 indicates moderate cognitive impairment, 20–24 indicates mild cognitive impairment, and 24–30 indicates no cognitive impairment [14].(8)General characteristics, such as gender, height (cm), weight (kg), body mass index (kg/m²), drinking, smoking, education level, subjective age of the elderly, suicidal ideation, and health-type factors, were obtained.

### 2.3. Data Analysis

All continuous variables in this study are expressed as standard deviation mean (SD), and categorical variables are expressed as percentages (%) in their respective groups. A normality test was performed, and the significance of Kolmogorov-Smirnov and Shapiro-Wilk was lower than the *p*-value of 0.05, so it was judged to be non-normal. The difference between all dependent variables, according to the presence or absence of driving, was verified using the Kruskal-Wallis test and the Chi-square test (frequency was 20.0% over performing a Fisher’s exact test). For the analysis, we used IBM SPSS Statistics for Windows, version 25.0 (IBM Corp., Armonk, NY, USA), and the statistical significance level was set at *p <* 0.05.

## 3. Results

The elderly who currently drive had a better subjective health status than those who did not. Among the current drivers, seven people had severe disabilities (grades 1–3), 44 had moderate disabilities (grades 4–6), 32 had physical disabilities, 11 had hearing impairments, three had visual impairments, and two had respiratory problems. At the time of the data investigation, most of the diseases had been cured, but there were differences between the groups in the treatment status of diabetes and chronic diseases, such as back pain, sciatica, pulmonary tuberculosis, and tuberculosis. The people who were not driving had more chronic diseases. In the currently driving group, the use of visual and hearing aids was 52.7% and 7.7%, respectively. Among the participants, 25.9% had discomfort due to bad eyesight, 15.1% had a hearing discomfort, and 28.0% experienced discomfort due to bending, squatting, kneeling, or reaching out for something higher than their heads. Of the respondents, 19.5% reported that it was difficult to perform touch movements. Depression symptoms decreased as they drove, and cognitive function was better in the driving group than in the other groups; however, it was also lower than the cut-off points for those over the age of 80. Among the elderly who were current drivers, 12.0% said that they experienced difficulties while driving in terms of decreased vision, hearing loss, decreased arm/leg reaction speed, decreased judgment (understanding of road conditions such as signals and intersections), and sense of speed. In other words, to prevent accidents due to aging, it is necessary to contribute to the safety management of elderly drivers by identifying their mental and physical conditions through precise identification of their mental and physical conditions.

### 3.1. General Characteristics

The general characteristics of the study participants were as follows: “current drivers” included 1729 men and 439 women; “past drivers but not current drivers” included 1237 men and 315 women; and 1045 men and 5225 women had “no driver’s license”. There was a difference between the groups with regard to age: “current drivers” 69.3(4.22), “past drivers but not current drivers” 74.08(5.74), and “no driver’s license” 74.58(6.54) (*p* < 0.001). Regarding the subjectively considered age of the elderly, there was a difference between the groups: 71.32(4.60) were “current drivers”, 69.72(4.14) were “past drivers but not current drivers”, and 70.02(4.04) had “no driver’s license” (*p* < 0.001). There was a difference in the presence or absence of disability determination as follows: 51 people were “current drivers”, 92 were “past drivers but not current drivers”, and 301 people had “no driver’s license” (*p* < 0.001). Regarding the degree of disability, “current drivers” comprised 7 people with severe disability (grades 1–3) and 44 people with moderate disability (grades 4–6); “past drivers but not current drivers” comprised 29 people with severe disability (grades 1–3) and 63 people with moderate disability (4–6); those with “no driver’s license” comprised 68 people with severe disability (1–3) and 233 people with moderate disability (4–6), exhibiting a group difference of *p* = 0.046. As for the usual subjective health status, 1598 people said they were “current drivers”, 749 people stated they were “past drivers but not current drivers”, and 2576 people stated they had “no driver’s license”; the perceived health difference was *p* < 0.001 (Table 1).

### 3.2. Current Disease Status and Their Treatment

The results of the current disease status and whether there were patients receiving treatment are as follows: although there were differences in most diseases, treatment was completed at the time of investigation; however, there was a difference between the groups in the presence or absence of treatment for diabetes (*p* = 0.01), musculoskeletal diseases (back pain, sciatica) (*p* < 0.001), and respiratory diseases (pulmonary tuberculosis, tuberculosis) (*p* = 0.037). The total number of chronic diseases diagnosed by doctors was 1.37 (1.24) for “current drivers”, 1.78 (1.50) for “past drivers but not current drivers”, and 2.02 (1.50) for “no driver’s license” exhibiting differences between the groups (*p* < 0.001). The number of prescription drugs being taken for more than 3 months was 1.31 (1.20) for “current drivers”, 1.78 (1.74) for “past drivers but not current drivers”, and 1.94 (1.55) for “no driver’s license” (*p* < 0.001) (Table 2).

### 3.3. Physical Function Status and Discomfort in Daily Life 

The following were the outcomes of the physical function status and discomfort in daily living: For those who answered “yes” regarding the use of a vision aid, 1142 people were “current drivers”, 890 people were “past drivers but not current drivers”, and 3247 people had “no driver’s license”; there was a difference between the groups (*p* < 0.001). As for those who answered “yes” in relation to the use of hearing aids, 1676 people were “current drivers”, 199 people were ” past drivers but not current drivers”, and 747 people had “no driver’s license”; there was a difference between the groups (*p* < 0.001). Those who were “uncomfortable” in their daily lives as a result of bad vision were as follows: “current drivers” consisted of 560 people, “past drivers but not current drivers” consisted of 508 people, and “no driver’s license” consisted of 2165 people; there was a difference between groups (*p* < 0.001). For discomfort due to hearing in daily life, “current drivers” consisted of 327 people, “past drivers but not current drivers” consisted of 383 people, and “no driver’s license” consisted of 1534 people who were “uncomfortable”; there was a difference between the groups (*p* < 0.001). Regarding the difficulty in performing motions (such as bending, squatting, or kneeling), “current drivers” consisted of 608 people, “past drivers but not current drivers” consisted of 770 people, and “no driver’s license” consisted of 3506 people who stated that it was “slightly or very difficult”; there was a difference between the groups (*p* < 0.001). For difficulty in performing movements (such as reaching out for something higher than their head), “current drivers” consisted of 423 people, “past drivers but not current drivers” consisted of 616 people, and “no driver’s license” consisted of 2911 people who stated that it was “slightly or very difficult”; there was a difference between groups (*p* < 0.001) (Table 3).

### 3.4. Depressive Symptom

As a result of examining the depressive symptoms, the score was 10.08 (2.21) for “current drivers”, 10.40 (2.20) for “past drivers but not current drivers”, and 10.34 (2.28) for “no driver’s license”, with a cut-off point of 8. The “current drivers” group exhibited a lower depression score than the “no driver’s license” (*p* < 0.001) group. Despite this, all groups were found to have high levels of depression.

### 3.5. Economic Activity 

The results related to economic activity were as follows: In relation to current economic activity, 1432 people were from the “current drivers” group, 448 people from the “past drivers but not current drivers” group, and 1898 people from the “no driver’s license” group were “currently working”. There were 676 “current drivers”, 1041 “past drivers but not current drivers”, and 3116 having “no driver’s license” who had “previously worked but not currently working”. The “never worked” people who were “current drivers” were 60 people, “previously a driver but not currently” were 63 people, and 1256 people had “no driver’s license”; there was a difference between the groups (*p* < 0.001). As for the participants who would like to work in the future, there were 804 people who “didn’t want to work” who were “current drivers” and 1006 people who had “no driver’s license”; 4298 people indicated wanting to “continue their current work” of which 1135 people were “current drivers” and 334 people had “no driver’s license”; 1339 people wanted to “continue with current job”, of which 82 people were “current drivers”, 53 people were “past drivers but not current drivers”, and 130 people had “no driver’s license. There were 141 “current drivers”, 130 “past drivers but not current drivers”, and 379“having no driver’s license”; there was a difference between groups (*p* < 0.001) (Table 4).

### 3.6. Recognition Function 

The results reflecting age and educational level that affect cognitive impairment are as follows: Looking at overall cognitive impairment, the elderly who were in the “current drivers” group had less precognitive impairment than the “past drivers but not current drivers” and “no driver’s license” groups. However, in the driving group, there were participants with lower than the recognition function cut-off points of 30 in the age group of 80 years or older (Table 5).

### 3.7. Current Drivers 

The degree of difficulty in driving was as follows: 24 people found it to be very difficult; 238 people stated that it was somewhat difficult; 352 people stated that it was just so; 859 people stated that it was not difficult at all; and 689 people stated that it was not at all. The difficulties experienced while driving were “eyesight impairment” in 236 people, “hearing impairment” in 22 people, “decreased reaction speed in arms and legs” in 82 people, “decreased judgment” (understanding road conditions such as intersections) in 151 people, and “slow speed” in 123 people.

## 4. Discussion and Conclusions

The data for this study were obtained from the health and welfare data portal of the Korea Institute for Health and Social Affairs to identify the physical and mental status of the elderly who are currently driving. A total of 9,990 people took part in the survey in 2020. Choi stated that elderly drivers experiencing difficulties adapting to changes in driving conditions are aware of the driving risks, including deterioration in sight and hearing [11]. It has been shown that many elderly drivers choose to drive despite the deterioration in their sight and hearing, which is a result of their natural aging and can cause serious accidents. Lee also stated that elderly drivers’ ability to adapt to driving situations is related to the risk of traffic accidents, which means that the physical health of the elderly is highly correlated with their driving performance [19].

Aging is natural, but the deterioration of vision inevitably increases the risk of accidents associated with driving; hence, elderly drivers must accurately recognize their mental and physical conditions. Health status is highly correlated with the safety perception of driving. If the elderly are rewarded for good health status, [5] they will drive more cautiously. Previous studies also reported that elderly drivers become distracted while driving owing to the increased auditory processing load, which increases the risk of driving accidents owing to increased driving speed variability [11,12]. It has been recognized that the driving risk increases when the elderly drive [11]. In addition, complications that can lead to accidents and, consequently, cause social problems are also important when psychotic or cognitive impairment occurs in elderly drivers [5,11]. In reality, it is impossible to unconditionally ban the elderly from driving, but in particular, the elderly who have vision and hearing impairments should receive driving assistance through orthoses and treatment.

It was reported that the elderly who currently drive had a better subjective health status than those who did not. Among the “current drivers”, seven people had severe disabilities (grades 1–3), 44 had moderate disabilities (grades 4–6), 32 had physical disabilities, 11 had hearing impairments, three had visual impairments, and two had respiratory problems. At the time of the data investigation, most of the current diseases had been cured, but there were differences between the groups in the treatment status of diabetes and chronic diseases such as back pain, sciatica, pulmonary tuberculosis, and tuberculosis. The number of chronic diseases increased, resulting in the elderly not driving. In addition, for 28.0% of the respondents, bending, squatting, and kneeling movements were difficult, and for 19.5%, reaching for something higher than their head was difficult. Depression symptoms decreased as they drove, and cognitive function was better in the driving group than in the other groups, but it was also lower than the cut-off point for those over the age of 80. Among the elderly who are currently drivers, 12.0% said that they experienced difficulties while driving in terms of decreased vision, hearing loss, decreased arm/leg reaction speed, decreased judgment (understanding of road conditions, such as signals and intersections), and decreased sense of speed. In a study by Choi, elderly drivers were found to take drugs for hypertension, diabetes, and hyperlipidemia [11]. Also, regarding the economic activity results of elderly drivers, there is a significant difference between groups according to current drivers, drivers who have driven in the past, and those without a driver’s license. This means that driving and economic activities are significantly correlated, and drivers have a strong correlation with economic activity. In this study, diseases such as diabetes, lower back pain, and sciatica were significantly different from those in the other groups. These results suggest that elderly drivers are unaware of medical conditions that can negatively affect their driving. The findings of this study can facilitate the safety management of elderly drivers by better understanding their mental and physical status.

This study has some limitations. The results must be interpreted with caution, as the findings do not represent all elderly drivers in the Republic of Korea. Further, the findings do not reflect the actual driving situation. In addition, it was impossible to directly discuss the risk of driving due to neurological symptoms.

## Figures and Tables

**Figure 1 healthcare-11-00563-f001:**
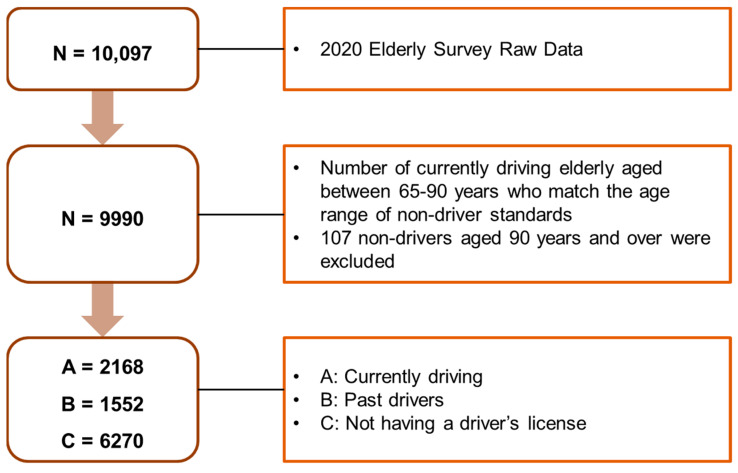
Data cleaning process flow.

**Table 1 healthcare-11-00563-t001:** General characteristics of the participants.

Characteristics		Driving							
		Current Drivers		Past But Not Current Drivers		No Driver’s License		X^2 3^/H ^4^	*p*-Value
		N ^5^/M ^1^	%/SD ^2^	N/M	%/SD	N/M	%/SD		
Sex	Man	1729	79.8	1237	79.7	1045	16.7	3864.248	<0.001
	Female	439	20.2	315	20.3	5225	83.3		
Height (cm)		167.68	6.77	166.22	6.92	157.29	7.13	3209.849	<0.001
Weight (kg)		66.74	7.56	65.02	7.78	58.29	7.98	1940.261	<0.001
Body mass index (BMI)		23.71	2.09	23.52	2.35	23.55	2.82	13.557	0.001
Age (years)		69.34	4.22	74.08	5.74	74.58	6.54	1192.218	<0.001
Recognition of elderly age criteria		71.32	4.60	69.72	4.14	70.02	4.04	167.758	<0.001
Education Level	Uneducated (not reading)	0	0	7	0.5	297	4.7	2320.532	<0.001
	Uneducated (reading)	10	0.5	50	3.2	739	11.8		
	Elementary school	261	12.0	394	25.4	2694	43.0		
	Middle school	483	22.3	431	27.8	1447	23.1		
	High school	1109	51.2	544	35.1	1013	16.2		
	College	126	5.8	41	2.6	36	0.6		
	University	179	8.3	85	5.5	44	0.7		
Disability	Yes	51	2.4	92	5.9	301	4.8	32.257	<0.001
	No	2117	97.6	1460	94.1	5969	95.2		
Degree of disability	Severe disability (1–3 degree)	7	13.7	29	31.5	68	22.6	6.154	0.046
	Moderate disability (4–6 degree)	44	86.3	63	68.5	233	77.4		
Disability type	Mental retardation	32	62.7	50	54.3	175	58.1	-	-
	Brain lesion disorder	1	2.0	8	8.7	19	6.3		
	Visual impairment	3	5.9	6	6.5	26	8.6		
	Deafness	11	21.6	16	17.4	51	16.9		
	Speech disorders	0	0.0	1	1.1	4	1.3		
	Intellectual disability	0	0.0	1	1.1	5	1.7		
	Autistic disorders	0	0.0	0	0.0	0	0.0		
	Mental disorders	0	0.0	1	1.1	6	2.0		
	Renal failure	1	2.0	2	2.2	6	2.0		
	Heart disorders	0	0.0	2	2.2	5	1.7		
	Respiratory disorders	2	3.9	3	3.3	1	0.3		
	Hepatic impairment	0	0.0	0	0.0	0	0.0		
	Facial disorders	0	0.0	1	1.1	0	0.0		
	Stoma disorder	1	2.0	1	1.1	2	0.7		
	Epilepsy disorder	0	0.0	0	0.0	1	0.3		
Regular exercise	Yes	1364	62.9	910	58.6	2951	47.1	191.752	<0.001
	No	804	37.1	642	41.4	3319	52.9		
Exercise time(min) / (1 time)		57	28	48	28	44	24	272.011	<0.001
Exercise frequency in 1 week	1 time	40	2.9	15	1.6	58	2.0	25.204	0.014
	2 times	141	10.3	85	9.3	242	8.2		
	3 times	287	21.0	207	22.7	595	20.2		
	4 times	102	7.5	63	6.9	222	7.5		
	5 times	370	27.1	212	23.3	794	26.9		
	6 times	93	6.8	86	9.5	270	9.1		
	7 times	331	24.3	242	26.6	770	26.1		
Smoking	Yes	514	23.7	278	17.9	310	4.9	666.616	<0.001
	No	1654	76.3	1274	82.1	5960	95.1		
Average amount of alcohol consumed (oz)		4.46	2.40	4.07	2.17	3.21	1.98	290.814	<0.001
Health status	Very healthy	247	11.4	54	3.5	131	2.1	889.457	<0.001
	Healthy	1351	62.5	695	45.6	2445	39.8		
	Normal	450	20.8	497	32.6	2139	34.8		
	Bad	113	5.2	234	15.4	1286	20.9		
	Very bad	1	0.0	43	2.8	145	2.4		

^1^ M: average, ^2^ SD: standard deviation, ^3^ X^2^: Chi-square test, ^4^ H: Kruskal-Wallis test, ^5^ N; frequency, *p*-value < 0.05.

**Table 2 healthcare-11-00563-t002:** Health status and health behavior.

Characteristics		Driving						X^2 3^/H ^4^	*p*-Value
		Current Drivers		Past But Not Current Drivers		No Driver’s License			
		N ^5^/M ^1^	%/SD ^2^	N/M	%/SD	N/M	%/SD		
Doctor’s diagnosis of hypertension	Yes	1134	52.3	899	57.9	3710	59.2	31.204	<0.001
	No	1034	47.7	653	42.1	2560	40.8		
Treatment of hypertension	Yes	1121	98.9	893	99.3	3657	98.6	3.518	0.172
	No	13	1.1	6	0.7	53	1.4		
Doctor’s diagnosis of stroke (Stroke, cerebral infarction)	Yes	37	1.7	81	5.2	295	4.7	41.999	<0.001
	No	2131	98.3	1471	94.8	5975	95.3		
Treatment of stroke (Stroke, cerebral infarction)	Yes	37	100.0	80	98.8	285	96.6	2.251	0.325
	No	0	0.0	1	1.2	10	3.4		
Doctor’s diagnosis of hyperlipidemia (dyslipidemia)	Yes	324	14.9	193	12.4	1188	18.9	46.082	<0.001
	No	1844	85.1	1359	87.6	5082	81.1		
Treatment of hyperlipidemia (dyslipidemia)	Yes	313	96.6	190	98.4	1164	98.0	2.662	0.264
	No	11	3.4	3	1.6	24	2.0		
Doctor’s diagnosis of angina pectoris and myocardial infarction	Yes	76	3.5	69	4.4	312	5.0	8.050	0.018
	No	2092	96.5	1483	95.6	5958	95.0		
Treatment of angina pectoris and myocardial infarction	Yes	74	97.4	67	97.1	306	98.1	0.335	0.846
	No	2	2.6	2	2.9	6	1.9		
Doctor’s diagnosis of heart diseases	Yes	65	3.0	63	4.1	329	5.2	19.785	<0.001
	No	2103	97.0	1489	95.9	5941	94.8		
Treatment of heart diseases	Yes	63	96.9	62	98.4	327	99.4	3.222	0.200
	No	2	3.1	1	1.6	2	0.6		
Doctor’s diagnosis of diabetes	Yes	421	19.4	401	25.8	1581	25.2	32.829	<0.001
	No	1747	80.6	1151	74.2	4689	74.8		
Treatment of diabetes	Yes	419	99.5	401	100.0	1557	98.5	8.644	0.013
	No	2	0.5	0	0.0	24	1.5		
Doctor’s diagnosis of thyroid disease	Yes	36	1.7	38	2.4	235	3.7	25.968	<0.001
	No	2132	98.3	1514	97.6	6035	96.3		
Treatment of thyroid disease	Yes	34	94.4	37	97.4	231	98.3	2.120	0.346
	No	2	5.6	1	2.6	4	1.7		
Doctor’s diagnosis of osteoarthritis (Degenerative arthritis)	Yes	140	6.5	143	9.2	1288	20.5	299.936	<0.001
	No	2028	93.5	1409	90.8	4982	79.5		
Treatment of osteoarthritis (Degenerative arthritis)	Yes	126	90.0	133	93.0	1193	92.6	1.318	0.517
	No	14	10.0	10	7.0	95	7.4		
Doctor’s diagnosis of osteoporosis	Yes	50	2.3	73	4.7	701	11.2	198.134	<0.001
	No	2118	97.7	1479	95.3	5569	88.8		
Treatment of osteoporosis	Yes	44	88.0	68	93.2	650	92.7	1.550	0.461
	No	6	12.0	5	6.8	51	7.3		
Doctor’s diagnosis of low back pain and sciatica	Yes	75	3.5	95	6.1	776	12.4	173.448	<0.001
	No	2093	96.5	1457	93.9	5494	87.6		
Treatment of low back pain and sciatica	Yes	63	84.0	88	92.6	651	83.9	5.048	0.080
	No	12	16.0	7	7.4	125	16.1		
Doctor’s diagnosis of fracture, dislocation, and aftereffects of accidents	Yes	17	0.8	19	1.2	93	1.5	6.242	0.044
	No	2151	99.2	1533	98.8	6177	98.5		
Treatment of fracture, dislocation, and aftereffects of accidents	Yes	15	88.2	16	84.2	83	89.2	0.390	0.823
	No	2	11.8	3	15.8	10	10.8		
Doctor’s diagnosis of fracture, chronic bronchitis, and emphysema	Yes	37	1.7	34	2.2	51	0.8	24.973	<0.001
	No	2131	98.3	1518	97.8	6219	99.2		
Treatment of chronic bronchitis and emphysema	Yes	34	91.9	33	97.1	45	88.2	2.111	0.348
	No	3	8.1	1	2.9	6	11.8		
Doctor’s diagnosis of asthma	Yes	19	0.9	40	2.6	116	1.9	16.151	<0.001
	No	2149	99.1	1512	97.4	6154	98.1		
Treatment of asthma	Yes	17	89.5	37	92.5	110	94.8	0.924	0.630
	No	2	10.5	3	7.5	6	5.2		
Doctor’s diagnosis of pulmonary tuberculosis	Yes	1	0.0	4	0.3	7	0.1	3.477	0.176
	No	2167	100.0	1548	99.7	6263	99.9		
Treatment of pulmonary tuberculosis	Yes	0	0.0	3	75.0	7	100.0	6.600	0.037
	No	1	100.0	1	25.0	0	0.0		
Doctor’s diagnosis of depression	Yes	6	0.3	22	1.4	113	1.8	26.942	<0.001
	No	2162	99.7	1530	98.6	6157	98.2		
Treatment of depression	Yes	6	100.0	18	81.8	100	88.5	1.634	0.442
	No	0	0.0	4	18.2	13	11.5		
Doctor’s diagnosis of dementia	Yes	8	0.4	27	1.7	137	2.2	31.401	<0.001
	No	2160	99.6	1525	98.3	6133	97.8		
Treatment of dementia	Yes	7	87.5	27	100.0	131	95.6	2.635	0.268
	No	1	12.5	0	0.0	6	4.4		
Doctor’s diagnosis of Parkinson’s disease	Yes	0	0.0	17	1.1	32	0.5	22.371	<0.001
	No	2168	100.0	1535	98.9	6238	99.5		
Treatment of Parkinson’s disease	Yes	0	0.0	17	100.0	32	100.0	-	-
	No	0	0.0	0	0.0	0	0.0		
Doctor’s diagnosis of insomnia	Yes	29	1.3	31	2.0	130	2.1	4.764	0.092
	No	2139	98.7	1521	98.0	6140	97.9		
Treatment of insomnia	Yes	22	75.9	25	80.6	106	81.5	0.488	0.784
	No	7	24.1	6	19.4	24	18.5		
Doctor’s diagnosis of cataract	Yes	93	4.3	70	4.5	282	4.5	0.177	0.915
	No	2075	95.7	1482	95.5	5988	95.5		
Treatment of cataract	Yes	78	83.9	57	81.4	205	72.7	6.008	0.050
	No	15	16.1	13	18.6	77	27.3		
Doctor’s diagnosis of glaucoma	Yes	18	0.8	21	1.4	50	0.8	4.465	0.107
	No	2150	99.2	1531	98.6	6220	99.2		
Treatment of glaucoma	Yes	14	77.8	20	95.2	40	80.0	2.914	0.233
	No	4	22.2	1	4.8	10	20.0		
Doctor’s diagnosis of chronic otitis media	Yes	16	0.7	13	0.8	27	0.4	5.261	0.072
	No	2152	99.3	1539	99.2	6243	99.6		
Treatment of chronic otitis media	Yes	16	100.0	13	100.0	26	96.3	1.094	0.579
	No	0	0.0	0	0.0	1	3.7		
Doctor’s diagnosis of senile deafness	Yes	15	0.7	48	3.1	146	2.3	30.050	<0.001
	No	2153	99.3	1504	96.9	6124	97.7		
Treatment of senile deafness	Yes	9	60.0	33	68.8	83	56.8	2.129	0.345
	No	6	40.0	15	31.3	63	43.2		
Doctor’s diagnosis of skin disease	Yes	23	1.1	15	1.0	29	0.5	11.072	0.004
	No	2145	98.9	1537	99.0	6241	99.5		
Treatment of skin disease	Yes	20	87.0	15	100.0	23	79.3	3.478	0.173
	No	3	13.0	0	0.0	6	20.7		
Doctor’s diagnosis of cancer (malignant neoplasm)	Yes	33	1.5	39	2.5	95	1.5	7.911	0.019
	No	2135	98.5	1513	97.5	6175	98.5		
Treatment of cancer (malignant neoplasm)	Yes	30	90.9	36	92.3	81	85.3	1.345	0.548
	No	3	9.1	3	7.7	14	14.7		
Doctor’s diagnosis of gastroduodenal ulcer	Yes	94	4.3	69	4.4	272	4.3	0.037	0.982
	No	2074	95.7	1483	95.6	5998	95.7		
Treatment of gastroduodenal ulcer	Yes	90	95.7	66	95.7	253	93.0	1.314	0.518
	No	4	4.3	3	4.3	19	7.0		
Doctor’s diagnosis of hepatitis	Yes	6	0.3	5	0.3	22	0.4	0.273	0.873
	No	2162	99.7	1547	99.7	6248	99.6		
Treatment of hepatitis	Yes	5	83.3	3	60.0	21	95.5	4.675	0.056
	No	1	16.7	2	40.0	1	4.5		
Doctor’s diagnosis of liver cirrhosis	Yes	5	0.2	11	0.7	15	0.2	9.434	0.009
	No	2163	99.8	1541	99.3	6255	99.8		
Treatment of liver cirrhosis	Yes	5	100.0	11	100.0	14	93.3	1.428	1.000
	No	0	0.0	0	0.0	1	6.7		
Doctor’s diagnosis of chronic kidney disease	Yes	9	0.4	30	1.9	55	0.9	23.091	<0.001
	No	2159	99.6	1522	98.1	6215	99.1		
Treatment of chronic kidney disease	Yes	9	100.0	28	93.3	54	98.2	1.663	0.472
	No	0	0.0	2	6.7	1	1.8		
Doctor’s diagnosis of prostatic hyperplasia	Yes	118	5.4	123	7.9	100	1.6	185.803	<0.001
	No	2050	94.6	1429	92.1	6170	98.4		
Treatment of prostatic hyperplasia	Yes	110	93.2	119	96.7	97	97.0	2.440	0.295
	No	8	6.8	4	3.3	3	3.0		
Doctor’s diagnosis of urinary incontinence	Yes	19	0.9	27	1.7	266	4.2	71.951	<0.001
	No	2149	99.1	1525	98.3	6004	95.8		
Treatment of urinary incontinence	Yes	9	47.4	18	66.7	125	47.0	3.812	0.419
	No	10	52.6	9	33.3	141	53.0		
Doctor’s diagnosis of anemia	Yes	13	0.6	23	1.5	93	1.5	10.392	0.006
	No	2155	99.4	1529	98.5	6177	98.5		
Treatment of anemia	Yes	10	76.9	22	95.7	76	81.7	3.279	0.175
	No	3	23.1	1	4.3	17	18.3		
Doctor’s diagnosis of ETC	Yes	40	1.8	24	1.5	128	2.0	1.704	0.426
	No	2128	98.2	1528	98.5	6142	98.0		
Treatment of ETC	Yes	36	90.0	24	100.0	122	95.3	2.670	0.273
	No	4	10.0	0	0.0	6	4.7		
Doctor’s diagnosis total number		1.37	1.24	1.78	1.50	2.02	1.50	356.311	<0.001
Prescription medication that currently taking for more than 3 months		1.31	1.20	1.78	1.74	1.94	1.55	315.923	<0.001

^1^ M: average, ^2^ SD: standard deviation, ^3^ X^2^: Chi-square test, ^4^ H: Kruskal-Wallis test, ^5^ N: frequency, *p*-value < 0.05.

**Table 3 healthcare-11-00563-t003:** Physical function and daily life discomfort.

Characteristics		Driving						X^2 3^/H ^4^	*p*-Value
		Current Drivers		Past But Not Current Drivers		No Driver’s License			
		N ^1^	% ^2^	N/M	%	N/M	%		
Assisted with eyesight	Yes	1142	52.7	890	57.3	3247	51.8	15.459	<0.001
	No	1026	47.3	662	42.7	3023	48.2		
Assisted with hearing	Yes	167	7.7	199	12.8	747	11.9	34.099	<0.001
	No	2001	92.3	1353	87.2	5523	88.1		
Assisted with chewing	Yes	530	24.4	558	36.0	2546	40.6	181.906	<0.001
	No	1638	75.6	994	64.0	3724	59.4		
Discomfort of eyesight	Not uncomfortable	1602	74.1	1015	66.6	3981	64.8	68.161	<0.001
	Uncomfortable	524	24.2	465	30.5	2039	33.2		
	Very uncomfortable	36	1.7	43	2.8	126	2.1		
Discomfort of hearing	Not uncomfortable	1835	84.9	1140	74.9	4612	75.0	97.336	<0.001
	Uncomfortable	308	14.2	343	22.5	1400	22.8		
	Very uncomfortable	19	0.9	40	2.6	134	2.2		
Discomfort of chewing	Not uncomfortable	1611	74.5	934	61.3	3608	58.7	173.696	<0.001
	Uncomfortable	501	23.2	522	34.3	2252	36.6		
	Very uncomfortable	50	2.3	67	4.4	286	4.7		
Muscle strength when sitting in a chair or bed and then getting up 5 times	Performed	2008	92.6	1126	72.6	4174	66.6	586.185	<0.001
	Tried but failed to perform (5 times not successful)	82	3.8	302	19.5	1594	25.4		
	Inability to even attempt to perform (elderly people with a vortex, or other disabilities that make it impossible to stand up)	10	0.5	43	2.8	174	2.8		
	Want to do it now	68	3.1	81	5.2	328	5.2		
Difficulty in performing movements such as jumping one lap (400 m) on the playground	Not difficult at all	576	26.6	189	12.2	454	7.2	1193.227	<0.001
	Slightly difficult	882	40.7	437	28.2	1371	21.9		
	Very difficult	508	23.4	580	37.4	2414	38.5		
	Cannot do it at all	163	7.5	304	19.6	1920	30.6		
	Do now	39	1.8	42	2.7	111	1.8		
Difficulty performing movements such as walking one lap (400 m) on the playground	Not difficult at all	1606	74.1	807	52.0	2493	39.8	826.431	<0.001
	Slightly difficult	414	19.1	477	30.7	2107	33.6		
	Very difficult	124	5.7	187	12.0	1157	18.5		
	Cannot do it at all	16	0.7	72	4.6	469	7.5		
	Do now	8	0.4	9	0.6	44	0.7		
Difficulty in climbing 10 steps without a break	Not difficult at all	1465	67.6	639	41.2	2030	32.4	907.291	<0.001
	Slightly difficult	550	25.4	567	36.5	2391	38.1		
	Very difficult	129	6.0	271	17.5	1415	22.6		
	Cannot do it at all	20	0.9	70	4.5	394	6.3		
	Do now	4	0.2	5	0.3	40	0.6		
Difficulty performing movements such as bending, squatting, or kneeling	Not difficult at all	1535	70.8	722	46.5	2449	39.1	682.021	<0.001
	Slightly difficult	482	22.2	551	35.5	2410	38.4		
	Very difficult	126	5.8	219	14.1	1096	17.5		
	Cannot do it at all	22	1.0	58	3.7	293	4.7		
	Do now	3	0.1	2	0.1	22	0.4		
Difficulty performing movements such as reaching out for something above the head	Not difficult at all	1729	79.8	895	57.7	3139	50.1	590.074	<0.001
	Slightly difficult	330	15.2	474	30.5	2162	34.5		
	Very difficult	93	4.3	142	9.1	749	11.9		
	Cannot do it at all	13	0.6	38	2.4	197	3.1		
	Do now	3	0.1	3	0.2	23	0.4		
Difficulty in performing operations such as lifting or moving about 8 kg of rice	Not difficult at all	1478	68.2	694	44.7	2097	33.4	855.233	<0.001
	Slightly difficult	496	22.9	519	33.4	2316	36.9		
	Very difficult	166	7.7	254	16.4	1304	20.8		
	Cannot do it at all	25	1.2	79	5.1	521	8.3		
	Do now	3	0.1	6	0.4	32	0.5		

^1^ N: frequency, ^2^ %: percentage, ^3^ X^2^: Chi-square test, ^4^ H: Kruskal-Wallis test, *p*-value < 0.05.

**Table 4 healthcare-11-00563-t004:** Social and economic activity.

Characteristics		Driving						X^2 3^/H ^4^	*p*-Value
		Current Drivers		Past But Not Current Drivers		No Driver’s License			
		N ^1^	% ^2^	N/M	%	N/M	%		
Current economic activity	Currently working	1432	66.1	448	28.9	1898	30.3	1305.474	<0.001
	Previously worked but not currently	676	31.2	1041	67.1	3116	49.7		
	Not working	60	2.8	63	4.1	1256	20.0		
Current work	Farmers and fisheries	353	24.7	77	17.2	482	25.4	854.529	<0.001
	Cost facilities management	159	11.1	92	20.5	132	7.0		
	Cleaning	59	4.1	65	14.5	468	24.7		
	Production	83	5.8	27	6.0	69	3.6		
	Household care	21	1.5	3	0.7	82	4.3		
	Driving transport	160	11.2	8	1.8	8	0.4		
	Professions	69	4.8	8	1.8	20	1.1		
	Office	37	2.6	7	1.6	8	0.4		
	Cooking and food	148	10.3	33	7.4	242	12.8		
	Courier and delivery	20	1.4	3	0.7	4	0.2		
	Site management	46	3.2	16	3.6	22	1.2		
	Environmental landscaping	27	1.9	27	6.0	114	6.0		
	Construction machinery	135	9.4	23	5.1	28	1.5		
	Culture and arts	9	0.6	0	0.0	3	0.2		
	Maintaining public order	14	1.0	16	3.6	69	3.6		
	Waste paper collection	5	0.3	8	1.8	24	1.3		
	ETC	87	6.1	35	7.8	123	6.5		
Work status	Don’t want to work	804	37.2	1006	66.1	4298	69.9	867.564	<0.001
	Continue with current job	1135	52.5	334	21.9	1339	21.8		
	Seeking different work	82	3.8	53	3.5	130	2.1		
	Do not work now, but want to work	141	6.5	130	8.5	379	6.2		

^1^ N: frequency, ^2^ %: percentage, ^3^ X^2^: Chi-square test, ^4^ H: Kruskal-Wallis test, *p*-value < 0.05.

**Table 5 healthcare-11-00563-t005:** Precognitive function (MMSE-K).

Characteristics			Education Level							
			0–3 Years		4–6 Years		7–12 Years		13 Years or More	
			Male	Female	Male	Female	Male	Female	Male	Female
Current drivers	Age	65–69	30 (2)	30 (1)	27 (73)	27 (40)	27 (776)	27 (272)	28 (165)	29 (39)
		70–74	30 (2)	25 (1)	26 (60)	25 (14)	26 (344)	27 (40)	29 (66)	28 (5)
		75–79	27 (2)	24 (1)	25 (45)	26 (7)	27 (120)	26 (9)	26 (16)	29 (2)
		80 over	22 (3)	16 (1)	23 (15)	23 (4)	27 (30)	6 (1)	28 (10)	30 (2)
Past but not current drivers		65–69	26 (1)	23 (1)	24 (42)	25 (16)	25 (185)	27 (126)	24 (14)	28 (15)
		70–74	21 (5)	25 (1)	25 (71)	26 (15)	25 (249)	27 (62)	26 (32)	27 (4)
		75–79	22 (11)	25 (4)	24 (117)	26 (20)	25 (211)	25 (19)	26 (29)	25 (6)
		80 over	20 (29)	15 (4)	23 (96)	23 (9)	24 (114)	24 (11)	25 (31)	29 (2)
No driver’s license		65–69	21 (6)	23 (41)	26 (36)	25 (385)	24 (141)	26 (1097)	26 (10)	26 (37)
		70–74	25 (9)	22 (132)	23 (94)	24 (602)	23 (147)	25 (513)	28 (8)	25 (16)
		75–79	23 (14)	21 (256)	23 (113)	23 (616)	23 (106)	24 (241)	27 (9)	24 (14)
		80 over	22 (72)	20 (514)	22 (180)	22 (627)	21 (91)	23 (129)	26 (9)	24 (5)

## Data Availability

Not applicable.

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
