# Peer review of "Health Status and Activity Discomfort among Elderly Drivers: Reality of Health Awareness"

_healthcare, 2023, doi:10.3390/healthcare11040563_

Round 1

Reviewer 1 Report

Dear Authors,

the topic of this paper seems interesting, but its form and style need to be severely improved. 

1. In lines 35-36 and 45-46 there is a repetition of the same concept.

2. In lines 81-82 three drivers groups are considered, while in lines 86-87 only two groups are described. 

3. In line 121 MMSE scores are defined wrongly. 

4. In paragraph 2.2, data variables should be listed in a bulleted list to be more intelligible. 

5. The results are presented in a confusing way, making them difficult to read.

6. It is not useful to show the test statistic in the results' tables without depicting critical values. 

7. "Disability type" variable shows too many categories and should be redefined in order to apply chi-squared test. 

8. Chi-squared test p-value is not applicable in the case of small cells of the contingency tables. Why Fisher exact test was not considered?

9. Data analysis is poor. It is not clear what normality test has been adopted (Shapiro-Wilk? Kolmogorov-Smirnov? Jarque-Bera?) and what the results were. 

Author Response

Once again, we are grateful that the reviewer acknowledged the improvements in the manuscript, and we thank the editor and the reviewer for their additional constructive comments and criticisms. We have revised our manuscript as per the recommendations of the reviewers.

Point-by-point responses to your comments and changes made when revising the manuscript are provided below. The revised manuscript has been submitted to the website.

Here is a point-by-point response to the reviewer's comments and concerns.

We hope this modification meets your approval.

Reviewer 2 Report

The authors in their paper analyzed the driving-related physical characteristics, underlying diseases, and health perceptions of elderly drivers in  Korea. the paper is scientifically sound. however many points should be discussed properly.

Do the authors really think the data set is sufficient to draw some conclusions from their study?

authors divided the set of data into many subsets however in the discussion authors didn't explain the relation of many subsets with their results.

are the conclusion can be valid for other countries or those conclusions only valid in Korea? If so, why does the reader from the title think that the study takes into account a set of data from many countries to get those conclusions?

  1.  

Author Response

Once again, we are grateful that the reviewer acknowledged the improvements in the manuscript, and we thank the editor and the reviewer for their additional very constructive comments and criticisms. We have revised our manuscript as per the recommendations of the reviewers.

Point-by-point responses to your comments and changes made when revising the manuscript are provided below. The revised manuscript has been submitted to the website.

Here is a point-by-point response to the reviewer's comments and concerns.

We hope this modification meets your approval.

Reviewer 3 Report

This is useful information but the analysis is quite simple.

So many characteristics affects driving status. I believe these characteristics also have correlations. 

If this data is not cross section but panel data analysis would be useful.

Author Response

Once again, we are grateful that the reviewer acknowledged the improvements in the manuscript, and we thank the editor and the reviewer for their additional constructive comments and criticisms. We have responded to the reviewers’ comments in detail below.

Q) This is useful information but the analysis is quite simple.
So many characteristics affects driving status. I believe these characteristics also have correlations. 
If this data is not cross section but panel data analysis would be useful.

R) Thank you for the comment. This is a survey of the elderly conducted by the Ministry of Health and Welfare, a government agency of the Republic of Korea, and its affiliated agency, the Korea Institute of Health and Social Affairs. It is a statutory investigation based on Article 5 of the Elderly Welfare Act, which has been conducted every three years since 2008 since the enactment of the law in 2007 and the fifth investigation in 2020. So, first of all, we are analyzing the trend of elderly drivers through the analysis of a single year in 2020. Furthermore, we are analyzing it to be submitted as a multi-year study in the form of a panel in 2023.

Round 2

Reviewer 1 Report

Dear Authors,

Thank you for your adjustments.

Still, I believe that some more changes are needed.

1) English is poor in the Abstract. You should provide synonims in order to variate the language (e.g. As the number of elderly drivers rapidly increases worldwide, interest in the dangers of 19 driving is increasing as accidents increase. The purpose of this study was to study the driving risk 20 factors of elderly drivers. In this study […]).

2) As previously said, MMSE scores are defined wrongly. In the paper you wrote:

“A total mini-mental state examination 353 (MMSE) score of 30 points is considered the cut-off point of cognitive impairment; a 354 score of 0–17 indicates no cognitive impairment, 18–23 indicates mild cognitive 355 impairment, and 24–30 indicates severe cognitive impairment [14].”

These scores are incorrect and described reversely. For a Korean reference, see Table 3 in Kang et al, 1997, “A validity study on the korean mini-mental state examination (K-MMSE) in dementia patients”: MMSE test is considered normal (NO cognitive impairment) when score is >= 24; severe dementia for scores <10; mild and moderate dementia are in between. This is the correct interpretation of MMSE scores and the error must be fixed.

Author Response

We would like to thank the reviewers for their constructive comments and helpful suggestions, upon which we have revised the paper. After the revision, we believe that the quality of the paper is now significantly improved. Below, we address the reviewer’s comments point by point.

We hope this modification meets your approval.

Reviewer 2 Report

the authors have taken into account all my comments, I recommend the publication of the article

Author Response

We thank the reviewer for helpful suggestions and constructive comments for revising the paper. After the revision, I think the paper's quality has improved considerably. 

Reviewer 3 Report

Please clearly describe the hypothesis.

Not the simple cross tabulation, but more sophisticated analysis should be applied.

Author Response

(The authors gave the same response as above.)
